# The Logic of Condom Use in Female Sex Workers in Bali, Indonesia

**DOI:** 10.3390/ijerph17051627

**Published:** 2020-03-03

**Authors:** Pande Putu Januraga, Julie Mooney-Somers, Hailay Abrha Gesesew, Paul R Ward

**Affiliations:** 1Department of Public Health and Preventive Medicine, Faculty of Medicine, Udayana University, Bali 80232, Indonesia; 2Discipline of Public Health, College of Medicine and Public Health, Flinders University, Adelaide 5042, Australia; 3School of Public Health, Faculty of Medicine, University of Sydney, Sydney 2006, Australia; 4Epidemiology, College of Health Sciences, Mekelle University, Mekelle 1871, Ethiopia

**Keywords:** identity, condom use, agency, sex workers

## Abstract

Studies on human immunodeficiency virus (HIV) prevention practices of female sex workers often examine the use of condom as a single behaviour: using or not using condom. This study explores typologies of the logic of condom use as part of exercising meaningful identities from female sex workers’ perspectives. We employed in-depth interviews with a purposely selected 35 female sex workers in Bali, Indonesia. Information from the in-depth interviews was analysed using thematic framework analysis to develop typologies of female sex workers’ experiences on the logic of condom use and its relation to the construction of identity. We identified two main logics for not using condom: the prioritising of financial stability and romantic relationships over condom use. The main logic for using condom was to protect their health in order to improve their future economic security. Embedded within these logics, women chose to practise agency and negotiate meaningful identities consistent with their ideals of being responsible mothers, successful migrant workers and loyal partners. Our study concluded that female sex workers had clear logics for both the use and non-use of condom with their clients, highlighting the rational nature of female sex workers decision making.

## 1. Introduction

Human immunodeficiency virus (HIV) has led to devastating epidemics in the overall population in general and most at-risk populations, such as injecting drug users and female sex workers, in particular [1]. Indonesia is no exception, where sexual transmission has become the primary driver for the development of the epidemic in recent years. The average nationwide HIV prevalence in Indonesia is 0.4% (0.1–3.5%) [2]. The burden of HIV in the nation is higher among the most at-risk populations (MARPS) including the Female Sex Workers (FSWs), men who have sex with men (MSM) and people who inject drugs [2].

Evidence shows that one of the groups that is enormously affected by the epidemic is FSWs. For example, a series of sero-surveys conducted by national and local government consistently reported a high prevalence of HIV infection among FSWs (5–20%), predominantly among brothel-based FSWs [3]. A review of sero-surveys data in Bali reported that HIV prevalence among brothel-based FSWs in Bali has increased from 0.62% in 2000 to 20.2% in 2010 [3]. The high prevalence of HIV and mobility of FSWs and their clients make the epidemic not only a local problem but relevant for the national epidemic. The high prevalence of HIV infection among FSWs in Bali has not been ignored, as HIV prevention programs have been running since 1994. These include programs monitoring women for sexually transmitted diseases (STIs) including HIV and those providing care and delivering behavioural change interventions to increase condom use through health education programs [4,5]. However, the epidemic among brothel-based FSWs has grown significantly and consistent condom use remains low [3]. Such inconsistency in condom use has been identified as being related to a lack of power by FSWs by UNAIDS [6].

A systematic review of literature in Asia [7] to synthetize factors that determined the ability of FSWs to negotiate condom use reported the following factors: individual level factors such as sex workers’ knowledge, perception and power; interpersonal and environmental level factors including dynamics with clients and peer-related factors; and other factors such as access resources, poverty, stigma, the legal barrier and the role of media. While able to identify different factors influencing sex workers ability to consistently negotiate condom, the review highlighted the possibility that broader social factors are intertwined and potentially inconsistent, suggesting the important of future research to be more solution-focused and contextually driven [7]. Meanwhile, international literature on condom use and sex work has strongly advocated for research to explore the roles of human agency, subjective meaning and local context in everyday decisions of sex work practices, including condom use [8]. Therefore, this paper focuses on the logic or reasons for using or not using condom, the contexts of when and why they use or do not use condom and to what extent these categories or logics are mutually exclusive.

### Exercising Identity and Condom Use

In exploring the roles of human agency in decisions around condom use, a number of studies have stressed the role of collective agency and identity [9,10] and social and peer networks [11,12]. Concepts of community development, mobilisation or empowerment have become popular approaches for developing social networks among marginalised groups, including FSWs [13]. The main aim of the approach is to raise consciousness about a community’s prosocial identities in ways that may help members of the network gain support to reinforce these identities and roles. In addition, they may work towards identifying and challenging barriers to address wider issues of security and support and specific issues related to HIV prevention behaviours (e.g., negotiating the use of condom, etc.) [14]. Evidence argues that an understanding of social networks is important for appreciating the micro-structural and environmental influences on HIV risk behaviours, social identities and norms [15].

These studies share a similar concept of social identity [16] in which FSWs are expected to share a similar identity or belong to the same category which makes them act as a group. Public health researchers using this concept have focused on examining how the broader social structures of the sex work setting impacts health behaviors of FSWs. However, this broader focus overlooks the internal dynamics, self- and identity-politics and rationalities which may also influence sex worker condom use practices.

We want to note that condom use is not merely an individual choice. Sex work in Indonesia (and in brothels) is illegal, and district police periodically enforce the law by arresting the FSWs or deporting them back to where they come from [17] as most of them are migrants. These conditions create a challenge to provide care and deliver behavioural change interventions to increase condom use through health education programs [18], and a dilemma for health authorities in providing health services and health promotion programs because they do not wish to be seen as supporting or legalising prostitution. The power imbalance due to the effect of gender and nature of service also affect consistent condom utilization. Given that men are dominant (patriarchal society), and since they are paying for the service, they are in a position of power to use or not to use a condom [19].

In our paper, we conceptualise a FSW as having the possibility of occupying multiple identity positions or identities, not simply as a sex worker within a sex workers community. These multiple identities may have competing or conflicting expectations for condom use behaviours. By using this lens, this paper provides a unique perspective in more fully understanding the construction of self-identities of FSWs and how these self-identities influence logics and decisions around condom use.

## 2. Materials and Methods

### 2.1. Study Setting

The study was conducted in Bali, which is an Indonesian province with a high prevalence of HIV infection among key at-risk populations, including FSWs. The majority of FSWs still work in brothels, which in Bahasa are called a *lokalisasi* or *lokasi* (in English, location), [20]. A *lokalisasi* consists of a number of houses (or in Bahasa, *wisma)*, where a pimp hosts a number of FSWs. Some FSWs both live and work in the brothels. During the study there were at least 10 *lokalisasi* operating in Bali’s two largest cities, Denpasar and Badung. Some of them, such as the Semawang area in Denpasar, have been known as a place for prostitution, even before Indonesia’s proclamation as an independent country in 1945.

Most FSWs in Bali migrated from neighbouring islands or provinces such as Java and Lombok; the majority come from Bali’s closest neighbour, East Java Province [21]. Since Bali is a tourist destination with high levels of infrastructure development, more than half of their clients are also migrant workers from Java [22].

### 2.2. Study Design and Population

Data presented in this paper are part of larger multi-method study conducted in 2013 to examine the association of social capital and HIV prevention practices of FSWs in Bali. The original study consisted of qualitative in-depth interviews and a cross-sectional survey. The present paper reports from in-depth interviews of 35 FSWs from Bali. The study target population was FSWs in 10 locations: Denpasar (9 locations) and Badung (1 location).

### 2.3. Participant Recruitment and Data Collection

Given the sensitives of undertaking research among sex workers in Bali where sex work is illegal, recruitment was facilitated by the *Yayasan Kerthi Praja* (YKP), a local non-governmental organisation (NGO) that has worked for several years with communities in the study’s locations. The YKP provided an assessment of the numbers of FSWs working in each location in Denpasar and Badung using a list compiled through their outreach work on 9 January 2013. At that time there were 809 FSWs in the 10 brothel locations.

We have described our qualitative study protocol in detail elsewhere [17]. Briefly, we recruited 35 FSWs in 2014 and YKP’s outreach workers observed and informally approached women in the brothel locations for their interest to join the study. Women who expressed their interest were then directed by these outreach workers to the first author for the informed consent procedure. The first author, who conducted the interviews, checked women’s eligibility, which included being at least 18 years of age, could speak in Bahasa Indonesia, and willing to participate in an in-depth interview. We developed a purposive sampling framework to get maximum variation sampling comprising women’s ages, time working at the location, participation as a member of any support group and number of women working in each brothel complex. Enrolled sex workers provided verbal informed consent for qualitative interviews. While participants could choose to be interviewed in their working room or in a private room at YKP’s office, all participants chose to be interviewed at YKP. Interviews explored FSWs’ accounts on life histories including mobility and migration experiences, networks and supports within FSW communities and HIV prevention practices.

### 2.4. Data Analysis

Data were analysed using framework analysis [23] to achieve the contextual and diagnostic goals of the qualitative phase by identifying the typology of the logics for condom use whereby the data were mapped across women’s experiences of condom use and one or more identified themes. This method provides a clear system by which to address research questions within a predesigned sample (FSWs in Bali) and within specific issues (identity and condom use), and hence it was considered to be appropriate.

We began our analysis by familiarising ourselves with the full verbatim and reading this several times and then generating initial codes. Then we developed a working analytic framework to produce core themes. Based on the chart of the core themes produced, the first author then conducted mapping across women’s experiences on condom use. With the help from second and third authors, the mapping results were then interpreted by developing associations between identity work and condom use; the focus of this activity was to find typologies of the logics of condom use [23]. To support the themes, we added illustrative quotes using pseudo names. We used NVivo 10 software program (QSR international Pty Ltd., Doncaster, Victoria, Australia, 2018) for data analysis.

### 2.5. Ethics

The Social and Behavioural Research Ethics Committee at Flinders University, South Australia (number 5913 SBREC) and the Institutional Review Board (IRB) of Yayasan Kerthi Praja, Bali, Indonesia (number 040/YKP-IRB/2012) approved the study’s protocols. We assured all study participants about the confidentiality of their information and that no identifying details would be shared and published. We received informed consent from the study participants before the interviews. We offered Rp.100.000 ($AUD 10) for the participants in recognition of the time they spent in interviews and to recognise their valuable contribution to this research. We also offered the participants a free condom on the day of the interview.

## 3. Results

Thirty-five (35) FSWs participated in the in-depth interviews. The majority of FSWs, 20 (57%), were aged between 25–40 years old. With regards to time as sex workers, 23 (66%) FSWs were senior and the remaining participants were newcomers. Furthermore, 11 (31%) of them have a partner who visited them regularly, 10 (29%) of them lived with their partner at the time of data collection, and the remaining were either divorced, never married, with no partner or had a husband but lived alone.

The findings focus on the logic of condom use and how these different logics were deployed as part of multiple identities. Women described their reasons for using or not using condom in terms of three fluid, overlapping types of logic. FSWs’ experiences related to condom use may involve elements of more than one logic of prioritising economic status, prioritising romantic relationships and/or prioritising health.

### 3.1. Prioritising Economic Security

Within this logic, using or not using condom was communicated as a strategy of negotiating identity towards survival and financial resources. The underpinning rationale for this logic by FSWs could be tracked to a sense of financial responsibility to their family (in particular, their parents) and a linked perceived need to be seen as a successful migrant worker. As previously discussed in the study setting, the majority of sex workers in Bali migrated from other islands in Indonesia (particularly Java), and the ultimate goal of working in Bali as a sex worker was to gain income in order to look after their family and to create savings for the future in order to stop sex work.

Within this logic of prioritising economic capital and constructing a new identity as a successful migrant worker, women stated that condom use was a difficult task to fulfil. This is in the context of sex without a condom attracting a much higher payment than sex with a condom. In addition, since FSWs had low trust in and perceived support by other sex workers in the same brothels, meaning that they could not trust other sex workers to refuse clients wanting sex without a condom [17], FSWs reported that they chose to accept sex without a condom when ‘big money’ was offered, thus increasing their vulnerability to HIV and other STIs.
Cantik:Q:“Eee, in your opinion, why don’t they care, why can’t you all agree and unite to refuse clients who don’t want to use condom?”A:“Well, if someone gives big money…”Q:“Big money?”A:“He-eh, if someone offers a lot of cash, yeah what could you do, no condom, okay. Just as simple as that.”

FSWs’ decisions to accept sex without a condom could be seen as a logic of ‘finance over health’ and a strategy of constructing an identity as a successful migrant worker. This identity construction mainly started through a sense of obligation to take care of their family welfare in their homeland, as for many FSWs the brothel was ‘a place for a moment’, a temporary place to gain money and generate income for their future life in their beloved homeland. The majority of women came to Bali after experiencing social and economic disadvantages; thus, returning as a successful migrant worker, able to financially support their families and invest in economic capital, could be seen as a self-recovery effort to return to their homeland—a sense of pride and success.

Furthermore, FSWs symbolised their self-recovery by possession of vehicles, property and jewellery, all of which they were proud of. Cantik, a young sex worker, described the importance of possessing economic capital for her social status back home. Interestingly, since she was able to send money back to her family quickly after moving to Bali, she assumed that her neighbours might know where the money actually came from. The Indonesian public views sex work as a moral problem and contrary to the religious order [24]; thus, as she also admitted, people have labelled her income from sex work as haram (Arabic term, in English, ill-gotten money). However, she claimed that wisely spending her economic resources for social benefits of the village community has helped her to attain a better social status. Besides, she claimed she was able to ‘buy’ anyone, including the village head, and thus avoid questions about her work, including moral judgements:
A:“I don’t care what they think, nothing happens, even from the village head. With money you could shut off any mouth, they won’t dare to question me.”
Q:“Anything with money?”
A:“He-eh, even the village head, he won’t question me too.”
Q:“Because of money?”
A:“The most important thing is money, just give them uang rokok (bribe).”

In Indonesia, uang rokok is widely used to refer to a relatively small amounts of money given to people in positions who illegally facilitate administration or business activities. By using term *uang rokok* (in English, cigarette money or bribe) to refer to the money she dispensed to people of influence, including the village administrator, Cantik symbolised her status as businessperson who could buy anyone, including the influential people. By doing so, she tried to protect her investment and social status from questions about her work, including moral judgements.

### 3.2. Securing a Partner or Wife Identity Over Condom Use

Within this second logic, non-use of condom was mainly practised in the context of romantic relationships. This strategy was used to negotiate a ‘partner/wife’ identity as demanded by a common social value. Moving to the sex work field in Bali, being isolated from the wider communities because of their sex worker status and facing the ongoing lack of trust and social support from other sex workers at brothels [17], many FSWs relied on romantic relationships for reliable social support. While stating that the support provided by their romantic partners was more related to general matters, such as taking them to brothels for work and cheering them up, within the harsh sex work environment [17] these types of support were significantly important. As Mira, a newcomer sex worker, and Ayu, a senior sex worker, described, even simple advice and support as well as gentle treatment from their romantic partners could make a significant difference to these women’s feelings.
Mira:A:“He always, you know… supports me, he advised me to avoid drinking, he said I am still young and have a future; thus, I should use my money wisely and keep saving. As simple as that…”
Ayu:A:“Just like a husband, he shows he cares. For example, when I worried a bit, he asked what happened, took me out to help me forget my problems. When I get back from working, he massages me gently, he spoils me.”Q:“Could you get that kind of attention from other sex workers?”A:“No, of course no.”

Furthermore, many of the sex workers viewed romantic relationship as also a strategy for the future, a new identity as a ‘normal wife’ and as an escape from their stigmatised status as sex workers. Indeed, we found some FSWs had taken their romantic partners back to their homeland to be introduced to their family as a ‘good candidate’ for a husband. Within this gendered and cultural context, these women identified themselves as a lover and a candidate as a loyal wife. Indeed, some participants reported the term of *suami-suamian* to refer to the steadier status of their romantic partners. The original word is *suami* which in English means husband, when repeated and with the suffix *–an*, the meaning is ‘a man who is treated like a husband’. An example of quotes explaining the goal of a future identity as a married woman is expressed by Mika, a senior sex worker, who spoke about her dream to have a ‘normal’ life:
Mika:“When I felt so low, I prayed to the God, I wish I could get someone nice that could permanently take me away from this job. I don’t want to work like this forever, my poor kids, they deserve normal money that doesn’t come from this job.”

The challenge for consistent condom use is that romantic relationships appear to involve riskier sex practices. For example, Suri, a senior sex worker who lived in the brothel, stated that she believed that her romantic partner did not have other concurrent sexual relationships, and thus, having sex without a condom would be safe. Instead, she expressed her fear that the risk of her job might cause harm to her romantic partner. She raised this with him, but he was not concerned for his own health, putting this in ‘God’s hands’. In numerous cases, FSWs with romantic partners argued that not using condom is viewed as an act of love, and the risk of being infected was considered inherent to the love itself. Using this logic, condom use with romantic partners was viewed as inappropriate or less important.
Suri:Q:“How do you feel about him?”A:“How do I feel? You know, if you love someone, happy, yes like that, I feel good hahaha.”Q:“You said love?”A:“Yes hahaha.”Q:“Love and can you love him and still use condom?”A:“No, I believe he won’t cheat on me, no way he will do that to me, I trust him.”A:“He-eh. You know, we trust each other, once, I told him about the risks of my job, and the possibility of being infected and he said—I am not afraid to die. I asked him to use condom and he asked why we should. He said—No, I won’t, whatever happens to you, I am not afraid, I won’t go back. Die now or tomorrow is just the same, that’s God’s business. He just surrenders and accepts all consequences.”

### 3.3. Rebuilding Positive Identities Towards Healthy Behaviours

The last type of logic for condom use is prioritising health, whereby the use of condom was undertaken largely for the benefit of personal health. This typology was mainly motivated by self-interest, and not necessarily making a wider contribution to the HIV prevention in the community.

In the two previous logics, we have shown how not using condom could be deployed in the process of identity construction as a successful migrant worker and a partner of a lover. Despite this, our findings indicate some participants were able to insist on condom use with clients or even partners by illuminating a positive identity as a healthy woman. An example of a success story comes from Misa, a senior and quite mobile sex worker who had worked in several brothels for more than two years in Bali. She admitted that in the past, she sacrificed condom use to achieve her goal to invest hastily in economic capital. However, her experiences witnessing some women becoming infected by HIV contributed to her logic to prioritise health over short-term financial gain.
Misa:A:“For the last two years I always use condom.”Q:“Before that?”A:“I didn’t fully understand the consequence of my choice, but now I am much older, I am getting mature and wise in deciding things. I am fully aware of the danger now. That’s why I use condom. I see many women who have been infected, I am afraid.”Q:“Did you take a test? When was your first test?”A:“Three months after I started work.”Q:“Three months, did you consistently use condom after your first test?”A:“No.”Q:“Why, didn’t they educate you before and after the first test?”A:“Yes, but I wasn’t so sure about the reasons, I think because of my target to get as much money as I could.”Q:“How about now? Do you still have that target?”A:“Yes, but my health is number one now. I am afraid, extremely afraid.”

Another example of a participant who reported ignorance when first entering sex work but subsequently realised the threat of long-term loss of her productivity if she became infected was Subi, a relatively young sex worker but with experience seeing other sex workers suffering from AIDS.
Subi:A:“I read so many pamphlets, flyers and books on HIV/AIDS, also I saw someone who became so skinny after being infected, then she died. I am so afraid; ohh the consequence is so bad.”“Many, many of them, they offered me more, they said 200 thousand without condom (normally FSWs received 50–100 thousand rupiah, equal to $AUD 5–10), but if I get HIV infection, that money wouldn’t be enough, for sure…”

Furthermore, the prioritisation of health is also related to their responsibility back home to support their family financially. Some participants who reported that they insist on condom use, including Misa, argued that their health was a valuable asset to guarantee their function to take care of their children’s future. In this context, these FSWs negotiated a positive identity as a healthy and responsible mother or member of the family. In this case, the gendered and cultural value of the mother responsibility perceived by participants has created a positive identity towards protection of HIV and other STIs.
Misa:Q:“Why do you think it is important to take care of your health?”A:“For my own good.”Q:“For your own benefit?”A:“For myself, my family, my children”Q:“So if you get sick…”A:“My children will suffer…”

However, since there is a high self-stigma to the sex worker characteristics and also the sex worker identity [17], this logic of prioritising health did not involve the term ‘sex worker’ as a social identity shared by participants. The logic of prioritising health is not related to their position as sex workers but merely their perceived obligation to stay healthy for the sake of their family wellness and their long-term future back at home.

So far, we have outlined three typologies of logic for using or not using condom in which different identities were constructed by FSWs. As we highlighted earlier, these typologies are not necessarily discrete and separate; the majority of participants exercise their agency towards different meaningful identities to negotiate the use of condom. For example, while developing positive identities towards healthy behaviours, Subi also constructed different identities that enabled her to sacrifice condom use with her boyfriend. Furthermore, when explaining about her self-efficacy to use condom consistently in the future, she indicated a sense of doubt by stating it would depend on the situation in the future. She used the word *kepepet* (in English, trapped, forced to turn) to describe a condition where she might have no option other than sacrificing the use of condom for desperately needed income. She indicated that recently she has been able to maintain a good economic position that enabled her to develop her interest in healthy behaviours. Here is a quote from Subi’s interview.
Subi:Q:“Please answer honestly, you said in the last year you always used condom?”A:“Yes, always.”Q:“Always?”A:“He-eh. Except with my boyfriend.”Q:“How about regular clients?”A:“Always.”Q:“Are you confident that you’ll have the ability or power to always use condom with your clients in the future?”A:“Yes, maybe…”Q:“Maybe?”A:“I should…maybe…”Q:“Maybe? Why? Why aren’t you so sure about that?”A:“How about if I was ‘kepepet’ (trapped)?”

## 4. Discussion

FSWs efforts in constructing meaningful identities towards condom use negotiations are complex and ongoing. The prioritisation of financial gain, romantic relationships, and protection of their health for the future were connected to their social position in the wider society [25]. Instead of negotiating a collective identity as sex worker, women chose to practice agency and negotiate identities meaningful to them as mothers, partners, daughters and successful migrants. Referring to their objectives to survive for family well-being and/or to create the foundation for a better life back home, these women re-worked their identities from poor girls or women or parents or abandoned widows into meaningful identities such as responsible mothers, successful migrant workers, and loyal partners.

Above all, the majority of participants’ cases illustrated the prioritisation of economic achievement for defining their success and position within the setting of their homelands and the setting of brothels in Bali. The notion of prioritising economic success or supporting family wellness is pertinent in the HIV prevention literature [26]. In struggles to fulfil responsibility to help their family’s economic situation, many women faced financial insecurity that have led them to compromise on condom use with clients [27].

However, this study provides evidence of the success of some women in using condom consistently in their efforts to prioritise their health related to their identities as healthy women and responsible mothers back home. Thus, the influence of these social identities and roles on the condom use of FSWs and the gendered and cultural identity of FSWs could help researchers and policy makers to develop specific interventions to increase consistent condom use. However, inequality, competitive environments and the lack of trust and supports perceived by sex workers impede their ability to disperse their success more widely. Women were not motivated to join networks of sex workers and were reluctant to accept the identity of being a sex worker because of a strong sense of self-stigma about sex work as a profession and sex workers’ moral characteristics [17]. This is consistent with other research into social identities of FSWs, where respondents were averse to being willingly associated with the sex worker identity [10].

The study findings on romantic relationships with partners align with other qualitative articles exploring the typology of sex worker relationships with clients [28,29]. These papers reported that when clients became lovers or romantic partners, the relationships developed on a foundation of love, trust, respect, emotional and often material support and thus increased sex workers’ risk of getting an HIV infection because of decisions not to use condom with each other. The findings from the current study showed that some respondents tried to pursue the cognitive aspects of social capital (trust and support) from their relationships with the partners. Sex workers once again viewed their ‘practices’ with their partner as a natural aspect of their roles as loyal partners. This illustrates how, in normal everyday situations, social interaction with partners was governed by unwritten rules. Bourdieu’s [30] views corroborated this by justifying the unintentional’ and ‘taken for granted’ nature of everyday practice. Participants symbolised themselves as a partner or wife of the partner (a man who should be treated like a husband) in these relationships, including ignoring the use of condom.

Therefore, if the prevention programs still portray FSWs as ‘powerless’ subjects [31] who are passive recipients of public health programs rather than active agents with capacities and agendas in their own right, the HIV prevention programs might fail to enhance consistent condom use significantly [32]. Hence, there is great need for public health programs to encourage FSWs to participate in community activities to develop collective action and solidarity to improve individual and community status [10].

As the study found FSWs in Bali were exercising self-identity over collective identity as sex workers, the prevention programs trying to enhance community participation possibly will be more effective if they facilitate opportunities for fulfilling sex workers’ needs and goals. From the analysis of findings, FSWs ultimate goal is to invest in economic capital, enabling them to build and maintain their family well-being and recover their prestige. Groups or networks of sex workers that benefit their members economically and fulfil their other priorities could be more attractive than groups that simply discuss health or risks related to HIV. However, this does not mean that health promotion programs should focus only on individual material benefits. Rather, this is the starting point enabling women to develop networks first, and then to incorporate critical reflection and health promotion messages in the activities of the networks. The messages could show how collective action for community change towards consistent condom use could benefit individual goals in the long term by maintaining their responsibility as mothers and helping the family well-being. In addition, economic development programs could help sex worker communities to maintain financial security and provide them with alternative livelihoods and hope for the future [33].

The study had some limitations. First, the primary author, male, conducted the interviews with the FSWs. This may have affected the process of data collection. However, the interviewer knows the local language, norms and concerns raised by the women in Bali and Indonesia. Second, due to the nature of the study design, qualitative approach, and the context of the setting (Bali is a tourist place), we may not draw adequate inference. However, we reached the point of saturation.

## 5. Conclusions

Our study identified the following core themes for using and not using condom: life history and sex work trajectories, structural and environmental conditions, the nature and scope of cognitive and structural social capital in sex worker communities, sex workers’ identities and social positions, psychosocial factors in HIV prevention practices, and HIV prevention practices. As people with multiple identities, e.g., as mothers, workers and partners, FSWs configured their decisions around condom use within this contextually contingent terrain. Findings from this study contributes to the critical approach to social capital and its relationship to the wider concept of forms of capital, social field and habitus in a study of social capital and FSWs’ condom use negotiations. The HIV prevention programs in Indonesia should be more adaptive to these different contexts and logics of condom use negotiations by FSWs. Future research with detailed qualitative and quantitative analyses is required to compare and examine the comprehensive interaction of social-structural factors within different groups of FSWs in Bali in relation to HIV prevention practices, particularly condom use negotiations.

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
