# Peer review of "The Logic of Condom Use in Female Sex Workers in Bali, Indonesia"

_ijerph, 2020, doi:10.3390/ijerph17051627_

Round 1

Reviewer 1 Report

Thank you for the opportunity to review “The Logic of condom use”. This paper provides really fascinating data from interviews with sex workers in Bali about their condom use. However, the paper is not ready for publication in its current form, and below I suggest a number of ways in which the authors may better contextualize and theorize their study.

Context

The focus on identity is interesting but it risks framing sex workers’ condom use as an individualized phenomenon. The authors note around line 68+ the role of social and other contextual factors in shaping condom use, and it would be helpful for the authors to discuss this more to emphasize that condom use is not merely an individual choice. Here it would be interesting to hear in a separate section about the legal status of prostitution in Indonesia (the authors note that it is brothel based, but is this legal?) and that many of the women who trade sex are migrants. How do these legal and migratory factors, among others (eg poverty), shape women’s vulnerabilities in sex work, particularly to HIV (as a result of pressure or other reasons to not use a condom)? There are extensive literatures on these topics (eg sex work and migration) that the authors could draw from to set this context.

Also, there was little discussion of the role of gender dynamics: oftentimes women sex workers are cast as “vectors of disease” and this paper risks reifying this notion at times. Yes, it is true that women who trade sex are at risk for acquiring and transmitting HIV, but what role do their presumably and predominantly male clients have in encouraging or rejecting condom use, especially since they are paying for the service and therefore are in a position of (gendered) power?

Theoretical frame

In section 1.1 the paper alludes to but does not precisely elaborate a theoretical framework. While it seems that “identity” is the main theoretical concept in the paper, it is not theorized and defined very clearly. There are extensive literatures on identity across a range of fields that may be relevant to this study, for example in sociology (eg Goffman in on stigma and identity), as well as in sexuality studies, public health etc. Given this, the authors need to state clearly and explain their theoretical framework, particularly as it relates to identity. And then in the conclusion they need to explain more clearly how their study contributes to these theories/understandings of identity more broadly.

Methods

Was there an incentive for conducting the interview? (eg money) Why/why not?

Also, what ethics protocols (IRB or otherwise) were followed specifically, eg consent forms?

At lines 124-125 the authors write “identifying the typology of the logics for condom use whereby the data 
were mapped across women’s experiences on condom use and one or more identified theme. 
” I don’t follow this – can the authors explain more how the data was mapped and from where the themes came (and what themes)?

Findings & Conclusion

I think the authors do an excellent job outlining the logics that underlie why women use condoms or not; however, the paper would be improved if these findings were connected more to a broader body of theory. To specify these connections, the authors could—in addition to outlining a theoretical frame earlier in the paper—better connect the findings to this and re-work the concluding discussion to be less summative and, instead, indicate 2-3 ways their findings enhance identity theory and studies of sex work more broadly. The paper could then conclude wit hdirections for future research.

Thank you for the opportunity to review this very interesting paper!

Author Response

Reviewer 1

Comments and Suggestions for Authors

Thank you for the opportunity to review “The Logic of condom use”. This paper provides really fascinating data from interviews with sex workers in Bali about their condom use. However, the paper is not ready for publication in its current form, and below I suggest a number of ways in which the authors may better contextualize and theorize their study.

Thank you so much for your valuable comments, we address them point-by-point here below.

Context

The focus on identity is interesting but it risks framing sex workers’ condom use as an individualized phenomenon. The authors note around line 68+ the role of social and other contextual factors in shaping condom use, and it would be helpful for the authors to discuss this more to emphasize that condom use is not merely an individual choice.

Thank you, we have added this as per your suggestions— please check page 3, lines 94-102.

Here it would be interesting to hear in a separate section about the legal status of prostitution in Indonesia (the authors note that it is brothel based, but is this legal?) and that many of the women who trade sex are migrants. How do these legal and migratory factors, among others (eg poverty), shape women’s vulnerabilities in sex work, particularly to HIV (as a result of pressure or other reasons to not use a condom)? There are extensive literatures on these topics (eg sex work and migration) that the authors could draw from to set this context.

Thank you, we have addressed this too— please check page 3, lines 94-100.

Also, there was little discussion of the role of gender dynamics: oftentimes women sex workers are cast as “vectors of disease” and this paper risks reifying this notion at times. Yes, it is true that women who trade sex are at risk for acquiring and transmitting HIV, but what role do their presumably and predominantly male clients have in encouraging or rejecting condom use, especially since they are paying for the service and therefore are in a position of (gendered) power?

Thank you again, we have addressed the effect of gender and patriarchalism— please check page 3, lines 100-102.

Theoretical frame

In section 1.1 the paper alludes to but does not precisely elaborate a theoretical framework. While it seems that “identity” is the main theoretical concept in the paper, it is not theorized and defined very clearly. There are extensive literatures on identity across a range of fields that may be relevant to this study, for example in sociology (eg Goffman in on stigma and identity), as well as in sexuality studies, public health etc. Given this, the authors need to state clearly and explain their theoretical framework, particularly as it relates to identity.

Thank you for your critical observation, we have elaborated more in this version as per your comments— please check page 2, lines 66-76.

And then in the conclusion they need to explain more clearly how their study contributes to these theories/understandings of identity more broadly.

Thank you for your valuable comment. We have added a concluding remark noting the added value of this study— please check page 11, lines 590-97.

Methods

Was there an incentive for conducting the interview? (eg money) Why/why not?

Thank you, yes, we provided them Rp100 in recognition of their time— please check page 4, line 181.

Also, what ethics protocols (IRB or otherwise) were followed specifically, eg consent forms?

Yes, we use the IRB ethics protocols- we have added the IRB ethical clearance number in this revised version. We also sought consent form from the study participants and support letter form the institutions— please check page 4, lines 176-183.

At lines 124-125 the authors write “identifying the typology of the logics for condom use whereby the data 
were mapped across women’s experiences on condom use and one or more identified theme. 
” I don’t follow this – can the authors explain more how the data was mapped and from where the themes came (and what themes)?

Thank you so much for this‑ please check page 4, lines 146-159. We applied thematic analysis, and conducted the following: ”We began our analysis though familiarising the full verbatim and read this several times, and then able to generate initial codes. Then after, we developed a working analytic framework to produce core themes. Based on the chart of the core themes produced, the first author then conducted mapping across women’s experiences on condom us. With the helps from second and third authors, the mapping results was then interpreted by developing associations between identity work and condom use, the focus of this activity was to find typologies of the logics of condom use. To support the themes, we added illustrative quotes using pseudo names.”

Findings & Conclusion

I think the authors do an excellent job outlining the logics that underlie why women use condoms or not; however, the paper would be improved if these findings were connected more to a broader body of theory. To specify these connections, the authors could—in addition to outlining a theoretical frame earlier in the paper—better connect the findings to this and re-work the concluding discussion to be less summative and, instead, indicate 2-3 ways their findings enhance identity theory and studies of sex work more broadly. The paper could then conclude with directions for future research.

Thank you for your critical observation. We have added these important points in the concluding remark. Plesae check page 11, lines 494-98.

Thank you for the opportunity to review this very interesting paper!

We thank you for contribution as well.

Reviewer 2 Report

Thank you for this very interesting paper I enjoyed reading. It provides valuable focus on a topic that remains current and important. I have some comments below.

INTRODUCTION

This provides a detailed overview of the topic, especially around the choices facing female sex workers in the use of condoms. I do note, however, that apart from one exception, all citations are older than 5 years (some MUCH older). This really needs bringing up to date – there will have been further studies and theoretical developments that are much more recent.

MATERIALS AND METHODS

These are described, and are consistent with a qualitative approach. It would be useful to know if computer software was used to analyse the data [I expect from 35 in-depth interviews there will be a significant amount of text].

I note that the study has ethical approval, and there was a consenting process in place.

RESULTS

These show some interesting findings – the three core themes that emerge are interesting, especially theme 2. I like the use of quotes for illustration. 

DISCUSSION/CONCLUSION

The discussion draws out the main findings and makes additional analytical points to illustrate their significance. As in the background section, I note that many of the citations are a number of years old – were there not more recent studies you could have cited? You make valid points about the economic imperatives for not using a condom, though I think a more intriguing finding is around the ‘romantic’ attachments. This may not have been revealed in other studies, so should be perhaps explored in more depth – what are the true implications here?

The limitations section highlights important factors – I do note that you acknowledge it was a male interviewer (which I consider a major limitation). In addition, you should add a note about the general limitations of a qualitative approach – that it’s based on self-reporting and individual beliefs, so may not be ‘true’ as such, but reflect respondent perceptions – and also that data were collected some time ago. You should recommend further research to address this and validate the findings.

Finally, the closing sentence – “The current HIV prevention programs in Indonesia should be more adaptive to these different contexts and logics of condom use negotiations by female sex.” This is fine, but as data were collected in 2013, has Indonesia done anything more recently to address this, and does your recommendation actually apply to 2019/2020? Whatever, you should reword and remove ‘the current’ and ‘more’ unless you’re sure the current situation (i.e. now) is the same as 2013.

Reviewer recommendations

  1. Bring the citations in the introduction (and throughout) up to date.
  2. State if computer software was used to aid analysis.
  3. Perhaps expand theme 2 in the discussion section, which is likely to be the most intriguing finding (themes 1 and 3 will have been reflected much more in other studies).
  4. Address the limitations section (see comments) and make recommendations for further research.
  5. Consider rewording the final sentences(s) – are health and HIV policies in Indonesia still the same as 2013?

Author Response

Reviewer 2

Comments and Suggestions for Authors

Thank you for this very interesting paper I enjoyed reading. It provides valuable focus on a topic that remains current and important. I have some comments below.

INTRODUCTION

This provides a detailed overview of the topic, especially around the choices facing female sex workers in the use of condoms. I do note, however, that apart from one exception, all citations are older than 5 years (some MUCH older). This really needs bringing up to date – there will have been further studies and theoretical developments that are much more recent.

Thank you very much we have updated the available references—check references. 

MATERIALS AND METHODS

These are described, and are consistent with a qualitative approach. It would be useful to know if computer software was used to analyse the data [I expect from 35 in-depth interviews there will be a significant amount of text].

Thank you, yes we have used NVivo 10 software program, and we have added this in the revised version. Please check page 4, lines 173.

I note that the study has ethical approval, and there was a consenting process in place.

Thank you

RESULTS

These show some interesting findings – the three core themes that emerge are interesting, especially theme 2. I like the use of quotes for illustration. 

Thank you.

DISCUSSION/CONCLUSION

The discussion draws out the main findings and makes additional analytical points to illustrate their significance. As in the background section, I note that many of the citations are a number of years old – were there not more recent studies you could have cited?

Thank you so much and we have updated so of the available citations.

You make valid points about the economic imperatives for not using a condom, though I think a more intriguing finding is around the ‘romantic’ attachments. This may not have been revealed in other studies, so should be perhaps explored in more depth – what are the true implications here?

Thank you. This is valid point and ‘the romantic relationship’ as a reason for not using condom is discussed in paragraph one, saying: “The prioritisation of financial gain, romantic relationships, and protection of their health for the future were connected to their social position in the wider society.” Please check page 4, lines 444-51.

The limitations section highlights important factors – I do note that you acknowledge it was a male interviewer (which I consider a major limitation). In addition, you should add a note about the general limitations of a qualitative approach – that it’s based on self-reporting and individual beliefs, so may not be ‘true’ as such, but reflect respondent perceptions – and also that data were collected some time ago. You should recommend further research to address this and validate the findings.

Thank you we added the limitation— “due to the nature of the study design, qualitative approach, and the context of the setting, Bali is a tourist place, we may not draw adequate inference. We also recommend a future research direction in the conclusion section: Future research with detailed qualitative and quantitative analyses is required to compare and examine the comprehensive interaction of social-structural factors within different groups of Female sex workers in Bali in relation to HIV prevention practices, particularly condom use negotiations. Please check page 11, lines 478-98.

Finally, the closing sentence – “The current HIV prevention programs in Indonesia should be more adaptive to these different contexts and logics of condom use negotiations by female sex.” This is fine, but as data were collected in 2013, has Indonesia done anything more recently to address this, and does your recommendation actually apply to 2019/2020? Whatever, you should reword and remove ‘the current’ and ‘more’ unless you’re sure the current situation (i.e. now) is the same as 2013.

Thank you, we deleted the word current. Please check page 11, lines 493.

Reviewer recommendations

  1. Bring the citations in the introduction (and throughout) up to date— we updated the most available ones.
  2. State if computer software was used to aid analysis—we have stated this and included in our revised version.
  3. Perhaps expand theme 2 in the discussion section, which is likely to be the most intriguing finding (themes 1 and 3 will have been reflected much more in other studies). We have added clarifications in the discussion.
  4. Address the limitations section (see comments) and make recommendations for further research—we added the limitation and recommendation.
  5. Consider rewording the final sentences(s) – are health and HIV policies in Indonesia still the same as 2013?— we deleted the word ‘current’.

Round 2

Reviewer 2 Report

Thank you for addressing my recommendations. I have no additional comments to make.